# Precision Agroecology

Hannah Duff *, Paul B. Hegedus , Sasha Loewen, Thomas Bass and Bruce D. Maxwell

Department of Land Resources and Environmental Sciences, Montana State University, Bozeman, MT 59717, USA; paulhegedus@montana.edu (P.B.H.); roydenloewen@montana.edu (S.L.); tmbass@montana.edu (T.B.); bmax@montana.edu (B.D.M.)
* Correspondence: hannahduff@montana.edu

**Abstract:** In response to global calls for sustainable food production, we identify two diverging paradigms to address the future of agriculture. We explore the possibility of uniting these two seemingly diverging paradigms of production-oriented and ecologically oriented agriculture in the form of *precision agroecology*. Merging precision agriculture technology and agroecological principles offers a unique array of solutions driven by data collection, experimentation, and decision support tools. We show how the synthesis of precision technology and agroecological principles results in a new agriculture that can be transformative by (1) reducing inputs with optimized prescriptions, (2) substituting sustainable inputs by using site-specific variable rate technology, (3) incorporating beneficial biodiversity into agroecosystems with precision conservation technology, (4) reconnecting producers and consumers through value-based food chains, and (5) building a just and equitable global food system informed by data-driven food policy. As a result, precision agroecology provides a unique opportunity to synthesize traditional knowledge and novel technology to transform food systems. In doing so, precision agroecology can offer solutions to agriculture's biggest challenges in achieving sustainability in a major state of global change.

**Keywords:** precision agriculture; agroecology; sustainable agriculture; sustainability transition; agricultural biodiversity; sustainable food systems

## 1. Introduction

Agriculture is both a major cause and potential solution for current environmental issues. Modern industrial agriculture has increased yields over time, but this has come at a staggering cost to the environment. Despite modern industrial agriculture contributing to environmental issues like nitrogen pollution, soil degradation and habitat destruction, enhanced information availability and analysis offered by the industry has the opportunity to solve, rather than perpetuate problems in agricultural sustainability. The future of agriculture should promote productive, economically viable, socially just, and environmentally sound agri-food systems [1]. We have known for decades that sustainable intensification of agricultural production is required to feed and nourish the world's growing population, and there are many avenues being pursued in this endeavor such as changes in land use management, closing organic yield gaps, and shifting diets [2,3]. From these pursuits we have identified two dominant paradigms that offer differing solutions to the problems of modern agriculture (Figure 1). The production-oriented paradigm imagines solutions based on productivity, technology, and optimized input management. When pushed to its furthest extreme, the fear of "big data", "agribusiness", and "robot agriculture" deters many stakeholders and practitioners from engaging in such industrialized agriculture solutions [4]. Alternatively, the countermovement of ecologically oriented agriculture endorses a more holistic style of ecologically based agriculture that focuses on long term sustainability, ecological solutions, and conservation practices [5,6]. Critics of the latter suggest these movements are fleeting, unproductive, and lack scientific evidence [7,8]. Despite diverging paradigms, we make the case that global calls for the transformation of

food systems will require both applications of technology and agroecological transformation to create productive and sustainable agri-food systems. Below we describe *precision agroecology* as the use of modern technological farm instrumentation and tools collectively called "precision agriculture" (PA) to accomplish the goals of agroecology.

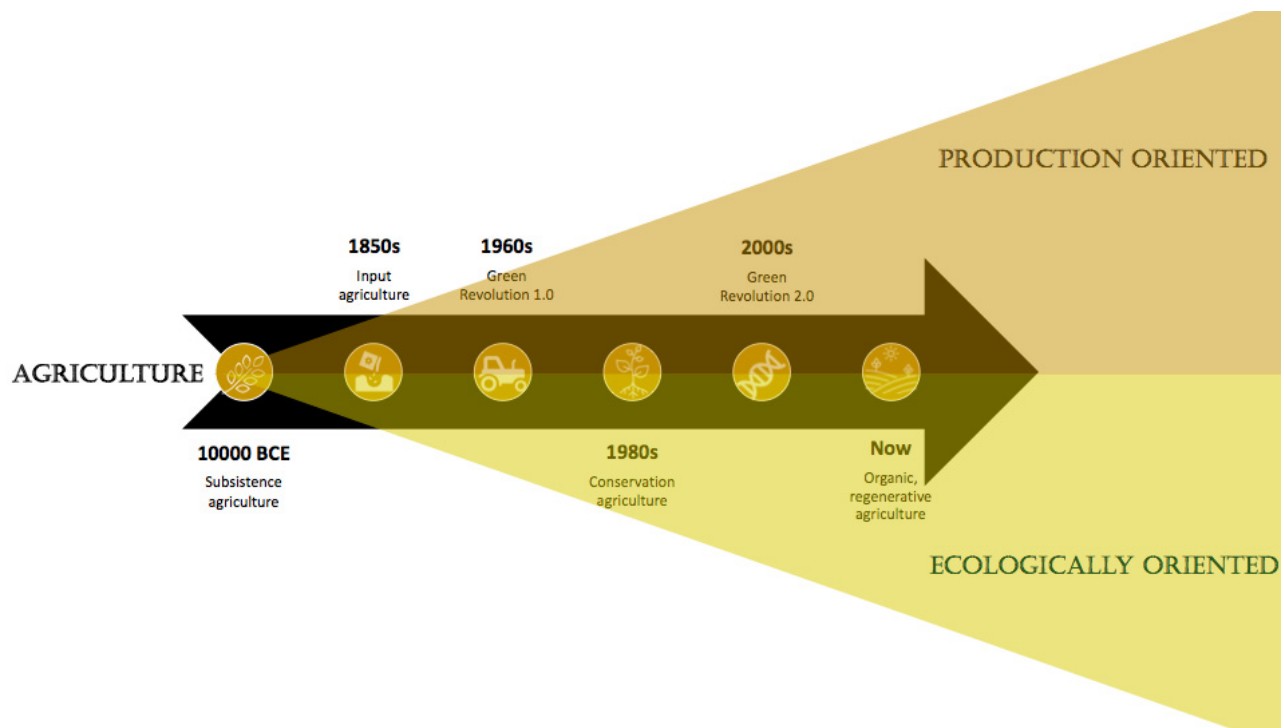

**Figure 1.** Depiction of the timeline of agriculture from subsistence agriculture to the present diverging paradigm of production-oriented and ecologically oriented agriculture.

Precision agriculture is often categorized within the production-oriented paradigm. Typically, PA is considered the collection of instruments that allow farmers to capture spatiotemporal data on their fields and apply it to management decisions that improve efficiency and quality, and thus sustainability of agriculture [9]. Rich data sets are being generated by farms every day and most farm machinery today collects, or at least interacts, with data in one or many ways [10]. The amount of farm information such as weather, topography, and vegetation indices available from machines, drones, weather stations, and satellite based remote sensing data is continuously increasing [11] and driving the adoption of big data analytics in agriculture [12–14]. Field specific data collection has been used to inform sub-field-scale management within fields by reducing generalizations made across spatial scales [15–19]. Greater exploitation of the data returned from daily precision farming operations can drastically increase resource use efficiency and produce crops in a manner that not only reduces the environmental impacts but increases the ecological and economic resilience of agroecosystems [20]. However, those who push back against PA lament the loss of stakeholder knowledge, data ownership, and values of small-scale farming. Many fear that PA will further distance producers from their land by substituting technology for local knowledge [21]. Others fear that it will prolong "productivist" values without regard for crop quality and encourage "ecological dystopias" [4]. In this sense, PA would only perpetuate the externalized costs of modern industrial agriculture on ecosystem and human health. As a consequence, PA may represent the industrial endpoint that not only encourages increased farm size but substitutes fully automated agriculture for human knowledge and land connection.

On the opposite side of the solutions spectrum, agroecology refers to a scientific discipline, an applied management practice, and a social movement [22]. The scientific discipline of agroecology is the study of the application of ecological concepts and principles to agroecosystems. Stemming from indigenous roots, agroecology goes beyond simply using ecological concepts to increase production and reduce environmental impacts, but uniquely adapts principles to communities based on the co-creation and sharing of knowledge, culture and food traditions, diversity, resilience, and responsible governance [23–25]. Agroecology looks beyond the farm level to the broader regional, national, and global levels of the food system to affect sustainable change. Agroecology also focuses on local knowledge, ensuring that farmers stay connected to their land and their management despite the arrival of technology and broad-based prescriptions [26,27]. Critics of agroecology point to economic realities and projected global food demand to belittle agroecology as nothing more than a local counterculture movement that is not capable of large-scale food production [28]. However, we propose the merger of these two movements by using PA technology to manage farms through specific application of agroecological principles.

We propose that PA and agroecology are compatible, rather than divergent strategies for creating sustainable agri-food systems. Although these two disciplines stem from seemingly incompatible backgrounds, they promote a sustainable agriculture that is profitable, equitable, minimizes environmental degradation and efficiently achieves these goals. Key to their integration is the idea that agroecosystems are complex and vary considerably over space and time. Application of agroecology has historically lacked adoption because it could only offer general principles (e.g., crop diversification) due to a lack of local field scale data that could account for the variability that results from reduced inputs and complex biological interactions. PA offers the local information required to make field-specific agroecological recommendations. PA is commonly misperceived as belonging solely to the "big tech" agribusiness world, but in reality it has roots in stakeholder-driven science and practice [29,30]. Precision agriculture technology and data are essential to create on-farm experimentation (OFE) and adaptive management [31]. OFE aims to formalize the research of farmers on their own fields [29]. Advances in technology, particularly in PA, have opened the door for farmers to formalize, detail, digitally record, and analyze their experiments in ways not possible before the agricultural data revolution [32]. Most farmers in industrialized nations have access to equipment that records inputs and harvest information; yet farmers and industry have not grasped the potential to use this data to manage fields in a site-specific manner. In addition to the massive stream of on-farm data from PA instruments and technology, there are large repositories of open-source satellite imagery that can be used to update management recommendations through adaptive management and apply on-farm experimentation. On-farm experimentation (OFE) is a collaborative form of science that synthesizes farmer's tacit knowledge and data to manage, improve, and even redesign agri-food systems [29,33]. These on-farm trials place agriculture in an ecological context that is site, history, and time specific.

At first glance, PA is an unusual ally of agroecology, as new technologies are most typically associated with conventional high-input agriculture. However, the initial goal of PA was "farming by soil" [34]. This means that precision tools allowed large-scale farmers to apply inputs across their fields specific and relevant to the soil types that were present. In this initial vision, farmers could spatially sample their fields and develop sub-field zones within which they could apply nutrient additions as needed. This form of thinking re-imagines the dominant paradigm wherein farmers apply average rates of inputs across entire fields and even entire farms in an effort to reduce the complexity of their operations. Using high input rates to overwhelm complex and variable natural systems is a hallmark of intensified agriculture. On the other hand, PA allows farmers to work with real field complexity using technology [35]. While Wendell Berry decried the use of farm technology, claiming it reduced a farmer's understanding of their fields [26], we can now see a future where technology can reintroduce a farmer to the complexity of the ecological interactions on their land [36].

Furthermore, PA has the potential to substitute data for synthetic inputs. Technologies of PA, including aerial maps and combine-mounted yield monitors, are used by producers and researchers alike to gain detailed site-specific (and free) data about agri-food systems. We argue that rather than distancing producers from their land, PA can reacquaint farmers with large fields and give them more extensive knowledge about the variation within their system [37]. This approach can help not only to improve farm management, but to mitigate environmental externalities by providing detailed quantitative analyses at the field and farm scale. For example, PA can reorient agricultural values away from pure production and facilitate "nutrition-sensitive agriculture" that prioritizes food quality by incentivizing producers and markets to manage and price crops for their quality rather than quantity [38]. Incentivizing producers to grow food for nutrient content and best management practices rather than net-return alone would align modern agriculture with the values of agroecology.

Although agroecology is commonly referred to as "a science, a movement and a practice" [22,39], the quantitative side of the discipline is often overlooked, and it should in fact be recognized as a mechanism to bring objective science to management, thus relieving the inefficiency of the trial-and-error approach. More commonly understood as a social movement, agroecology indeed emerged as a potential solution to the crisis of industrial agriculture in tandem with the environmental movement. Agroecological farming practices of the 1980s were well aligned with social movements of the 1990s that called for radical transformation of agriculture [40]. The goal was to remedy the social, economic and environmental externalities of agro-industry with an alternative agriculture movement [41]. Today, agroecology is a transdisciplinary concept that encompasses ecological, social and economic dimensions of food systems [42]. From a scientific perspective, agroecology draws from ecological principles and applies them to manage agricultural systems [43,44]. This includes concepts of evolutionary fitness, competition dynamics, and plant population modeling not commonly considered by traditional agronomists or producers [45–49]. Agroecology also values an ecological systems approach that considers production impacts at multiple spatial scales.

We propose that PA and agroecology are an unlikely, yet necessary pair for creating sustainable agricultural solutions. Precision agroecology is grounded in an ecological framework but employs the benefits of modern technology and data-intensive management from PA to monitor beyond the plot scale [50]. The fusion of PA and agroecology offers transformative agri-food systems solutions that were not previously possible. Therefore, we explore the potential of precision agroecology as the future of agriculture.

## 2. Five Tiers of Precision Agroecology

Uniting these two disciplines as precision agroecology offers a unique array of agroecological solutions driven by data collection, experimentation, and decision support tools. Stephen Gliessman, in his textbook on Agroecology, criticized yield maximization as the main goal of industrial agriculture and instead prioritized healthy ecosystem function as the foundation of food production [46]. Meeting current food requirements and protecting soil and water for future agricultural demand are vital [3], however, sustainable agriculture also requires reducing inequalities in food systems from the local to global scale. Sustainable agriculture requires collaboration and cohesion at all levels of food systems, from producers, processors, distributors, retailers, consumers, and the political entities that operate at each scale. We refer to Gliessman's [51] proposed five levels of transformation as a framework to convert conventional industrial food systems to agroecological systems using precision technology. Precision agroecology facilitates the first four levels through (1) increasing agrochemical efficiency, (2) substituting more sustainable inputs, (3) maximizing ecosystem services, and (4) reestablishing consumer–producer connections. The fifth level moves beyond the control of PA and calls for (5) creating a just and equitable global food system [51]. Within each agroecological tier, we show how precision agriculture (PA) technology can be used to execute agroecological concepts and enhance agri-food system

sustainability (Table 1). Throughout the text, we provide examples of how components of both PA and agroecology can be merged into precision agroecology practices at each tier of sustainable transformation. While this paper provides case studies of tiers one through four, we propose potential solutions for merging PA technology and agroecology in regard to tier five in the discussion section.

**Table 1.** Precision Agroecology Framework.

| Tiers of Agroecological Transformation | Precision Agriculture Component | Agroecology Component |
|---|---|---|
| Tier one: reduce inputs | Create optimized prescriptions for site-specific nitrogen fertilizer/manure/cover crop application | Reduce environmentally damaging inputs and externalities |
| Tier two: substitute sustainable inputs | Use variable rate technology to optimize cash crop, cover crop, and animal manure application rates | Replace environmentally damaging input rates with renewable, sustainable, site-specific ones |
| Tier three: redesign agricultural systems to incorporate biodiversity | Use yield maps and remote sensing to monitor beneficial ecosystem services from non-crop habitat | Increase biodiversity to increase ecosystem resilience and ecosystem services |
| Tier four: reconnect producers and consumers | Optimize values-based supply chains via production and transportation data | Form alternative food networks that are based on direct relationships |
| Tier five: create a just and equitable global food system | Utilize the data stream associated with PA to inform policy at all levels of agricultural and food systems | Account for the environmental and societal relationships surrounding agriculture and food systems |

### 3. Tier One: Reduce Inputs

The first agroecological tier calls for increasing efficiencies of applied agrochemical inputs used in modern conventional agriculture to maximize crop production. Elliot and Cole [52] recognized that tradeoffs between maximization of production and minimization of pollution were inevitable and called for the shift towards optimization of profits and sustainability in agricultural production. Gliessman's [46] first step moves conventional modern agriculture away from inefficient practices such as uniform applications of fertilizer and pesticides towards site-specific approaches that optimize production and increase economic and environmental sustainability of agricultural communities.

Site-specific management is an application of PA that can increase efficiency and address the issues surrounding excess agrochemical input rates. Precision agriculture accomplishes this by reducing input rates in areas where the crop response does not result in increased net returns. A common input for which site-specific management is utilized is nitrogen fertilizer. Diverting resources from low profit potential areas into high profit potential areas has two major results: in most cases reduction of total nitrogen applied over a field and less expenditure by producers on fertilizer [53,54]. Site-specific nitrogen management varies greatly in the methods used to develop prescriptions and the scale at which management units are applied [53–58]. Site-specific fertilizer applications have been investigated in diverse crop systems throughout the US [59–63] and profit maximizing site-specific nitrogen management has been shown to increase net returns in the wheat belt from Oklahoma to Montana [16,59]. Reducing nitrogen fertilizer contributes to the sustainability of the natural resource base that agriculture relies on and improves farmer net returns by increasing the efficiency of fertilizer applications. This puts more money in the pockets of producers and thereby the broader rural community.

Site-specific management of inputs is best optimized through OFE applied to crop production, to intentionally understand crop responses to variable rate application management. Site-specific management and OFE both require harnessing the stream of data gathered on farms and from remotely sensed data sources to power analyses and augment decision making. The automatic collection of data from farm machines is becoming easier through cloud software such as "MyJohnDeere" and from satellite image repositories

such as Google Earth Engine [64]. The spatiotemporal availability of remotely sensed data allows for enrichment of any on-farm dataset by, for example, providing information from remotely sensed weather estimates or topographical variables at the sub-field scale locations of harvest data points. Statistical and machine learning approaches can be used to characterize the response of the crop, in terms of production (yield) or quality (e.g., grain protein content), to variable nitrogen (N) fertilizer inputs and other environmental covariates. These models can then be used to simulate outcomes of various complex management approaches where farmers are provided with an array of management options that they can choose from, while ultimately leaving decision making in the hands of the farmer.

Current decision support systems have mainly been developed with models focused on profit maximization and have shown promise not only to increase farmer net returns but to minimize the amounts of chemical inputs within fields. Future work will be development of models optimized on maximizing profits and minimizing pollution, driven by OFE. Using our precision agroecological approach, site-specific optimization of competing goals can apply an agroecological lens to harness the power of PA and address issues of economic and environmental sustainability. Increasing chemical efficiency serves as the initial steppingstone for the transformation of industrial agriculture towards an agroecological framework but must not be an endpoint where agroecology is conformed to current agricultural practices [25]. Early conceptualization of agroecology envisioned the substitution of industrial synthetic inputs with information about ecological interactions. We now have the data availability to realize that substitution.

## 4. Tier Two: Substitute Sustainable Inputs

The second agroecological tier calls for substituting organic inputs, or knowledge, for industrial synthetic inputs. Organic agricultural systems have been attempting this at scale for decades at least, and PA can be an important tool in efficiently shifting towards more sustainable inputs [65]. As noted in tier one, synthetic nitrogen is one of the most ecologically damaging industrial agricultural inputs, alongside pesticides. Broadly, organic agriculture removes chemical inputs from the agroecological environment by substituting synthetic inputs with animal manure, cover crops, and local knowledge [66,67] This practice of substitution produces healthier food and reduces nonpoint agricultural pollution [68]. Animal manure is rich in nitrogen but is unavailable in many locations in North America. Cover crops, which include nitrogen fixing plants such as peas and hairy vetch, provide nitrogen where animal manure access is limited. Additionally, these crops can reduce weed pressure through competition and varied termination methods [69]. Organic farmers, faced with diverse challenges, rely on local knowledge to apply inputs with greater precision and timing than conventional farmers. The emphasis on understanding local conditions is greater in organic systems as they do not rely on pesticide options to manage pest outbreaks or synthetic fertilizers to correct low soil fertility. This notion of farming with local knowledge is something all farmers do, but organic farmers in particular tend to be systems thinkers who seek out new information to aid whole-farm planning and decision making [70]. Thus, they are well suited to add precision agricultural data management to their tool kit.

The primary drawback of organic agriculture is reduced yield outputs due to nitrogen deficiencies and weed pressures. However, PA and OFE can help close this yield gap [71]. Organic farmers can use OFE to rapidly understand the patterns of spatial and temporal variation across their fields and thus manage them more efficiently. Seeding rates of cash crops and cover crops impact crop quality, yield, and competitive ability [72–75]. Subsequently, organic OFE methodology focuses on applying experimental randomized seeding rates across entire fields to find optimum site-specific seeding rates. This methodology is applied to both green manure nitrogen fixing cover crops, and cash crops like wheat or hemp, in order to minimize weed pressure, optimize yields, and maximize farmer net-return. Beyond the yield maps and other topographic variables mentioned in tier one, weed survey maps can also be incorporated into models to reveal best management

practices. Early results from organic OFE research have revealed new spatially varied optimum seeding rates which outcompete farmer chosen uniformly applied whole field seeding rates. The farmer can choose to site-specifically optimize seeding rates to maximize profits and minimize nitrogen losses and the knowledge gained through OFE complements the farmer's historic knowledge of a field. Through OFE, an organic farmer can speed the process of understanding their land and the impact organic inputs have on outcomes such as yield and weeds. Increased local knowledge helps an organic farmer manage their land without the use of synthetic inputs, thereby enabling PA tools to enable sustainable transition from ecologically damaging inputs to organic ones.

## 5. Tier Three: Incorporate Diversity

The third tier of agroecological transformation entails redesigning agri-food systems to incorporate more diversity in ecosystem structure and facilitate ecological function [51]. Simplified agricultural systems are criticized as "ecological sacrifice zones" that disrupt ecosystems [35]. In contrast, diverse agroecosystems that conserve natural ecosystem structure have more complex ecosystem function. As a consequence, they provide many more ecosystem services that benefit producers in agricultural landscapes. Beneficial ecosystem services associated with biodiversity include pollination, pest predation, and weed seed predation [76–80], though tradeoffs may include increased pest habitat, increased weed density, and yield reduction [81]. In theory, agroecological principles such as diverse crop rotations, high biomass cropping systems and soil fertility building are key to maximizing ecosystem services in agri-food systems [49]. Plant diversity plays an important role in ecosystems and agroecosystems alike by enhancing ecosystem structure and function. Associated ecosystem services of higher plant diversity include enhanced nutrient cycling, soil quality, and habitat for beneficial insects [77,82,83]. In turn, these ecosystem services may provide agronomic benefits such as lower input costs, higher nutritional content in crops, and maintained or increased crop yields [84–86]. However, ecosystem services are notoriously difficult to quantify and monitor in ecological systems, making them extremely difficult for producers to manage [85,87]. We propose that site-specific, quantitative data from PA technology can be used as an on-farm conservation tool to optimize ecosystem services and manage tradeoffs in agricultural systems [88].

Precision conservation is facilitated by PA and can aid a transformation towards diverse agroecosystems [89,90]. Precision conservation accounts for spatial and temporal variability by using a suite of spatial variables to manage natural and agricultural systems [91]. In agricultural settings, precision conservation uses profit mapping technology to identify low-producing areas to create non-crop habitat in agricultural landscapes [76]. While most on-farm conservation efforts have focused on planned biodiversity, habitat management, and remnant habitats such as buffer zones and roadside margins, a broader category of ecological refugia can function as in-field precision conservation areas. Ecological refugia are uncropped patches in fields that serve as patch habitat to harbor biodiversity, beneficial insects and provide ecosystem services for producers [83,92,93]. Ecological refugia may be naturally occurring areas of terrain that are too difficult to cultivate or low-producing areas that are intentionally treated for restoration. In practice, ecological refugia can range from uncultivated riparian areas and rocky patches to intentionally planted patches of cover crops or pollinator strips.

Quantifying the economic and ecological effects of refugia is essential to producer adoption of this potential conservation practice in agricultural systems. Refugia must show an economic benefit in terms of crop production and ecological benefit in terms of biodiversity. Profit maps are an effective farm management tool that can be easily generated by PA technology. Annual profit maps can be used to monitor the effects of ecological refugia on crop production by quantifying crop yield and protein content as a function of distance from refugia. Producers may see the effects of beneficial ecosystem services via higher crop yields or nutrient content near the refugia compared to other locations in the field. Furthermore, precision conservation can save farmer's time and money by

taking low-yielding areas out of production. Ideally, this would increase their return on investment while increasing patch habitat and ecosystem services across the agricultural landscape [85]. At present, biodiversity surveys are typically required to quantify plant, insect and small mammal diversity surrounding the refugia, as remotely sensed data lacks the level of detail required for species-specific identification. However, recent developments in entomological lidar have made it possible to remotely monitor insect populations and activity using sensors to assess insect wingbeat frequency, color and wing to body ratio [94]. In addition, near-infrared spectroscopy can now accurately identify sagebrush up to the species (75–96%) and subspecies (99%) level, with vast implications for remotely monitoring vegetation at larger spatial and temporal scales [95].

Precision agroecology can merge PA data and agroecological principles to enhance the diversity of ecosystem structure and function in production systems. Agroecological concepts of biodiversity, ecosystem stability and ecosystem function can be monitored with precision technology and improved through agroecological management. Thus, PA's burgeoning technology and field automated data collection can augment efforts to assess if ecological refugia support biodiversity, enhance ecosystem services, or increase food production and quality. In this way, precision agroecology will reduce barriers to adoption and provide the tools needed for producers to participate in agri-environment schemes that offer payments to incorporate biodiversity into farmscapes [96].

## 6. Tier Four: Reestablish Consumer–Producer Relationship

Gliessman's [51] charge for tier four of food system transformation suggests reestablishing a more direct connection between those who grow our food and those who consume it. This goal is exemplified by growing demand for local food, both in terms of consumer interest and entrepreneurial activity. Local food sales were estimated at $4.8 billion in 2008 and $6.1 billion in 2012 [97,98], with subsequent iterations of these reports likely to show continued growth. To answer the charge, producer–consumer relationships must be restored by strengthening local/regional food systems (LRFSs) and fostering "food citizenship" on a large scale.

In contrast to traditional agricultural supply chains, an LRFS is better described as a values-based supply chain that aims to enhance producer profitability by paying price premiums for the environmental and social values implicit in their products [99]. Therefore, values-based supply chains require a high level of transparency and information sharing at each stage of the supply chain [99]. In this regard, values-based supply chains foster a food system that compensates producers for food quality, rewards best management practices, and relies on open accessible data flows to relay information to consumers. Fortunately, PA technology generates ample data that is free and site-specific to producers, that could be made readily available for consumers. This data holds the potential to transform value-based supply chains by offering evidence of producer practices and food nutritive quality that consumers are willing to pay for when made explicit. For instance, consumers have been found to be both "quality-focused" and "price-sensitive" in their willingness to pay when provided with traceable codes relaying information on food safety and quality [100]. By scaling up transparent data flow and traceable food choices, evaluations of consumer purchasing behavior can illuminate consumer's attitudes towards food nutrition and quality [101]. Accordingly, PA data flow can be scaled up to increase traceability, for example by using QR codes as labels to convey detailed information on production practices. Alternatively, data flow can be scaled down, for example many producers now use the Square app to interact with consumers face to face in small, local markets. In this sense, at scales both large and small, data-intensive labeling and software applications are reconnecting producers and consumers.

In contrast to conventional food systems, characterized by large-scale production, vertical integration and rigid controls of inputs and environmental variables, LRFSs are more embedded in the ecology and social structures of their location. The participating businesses and consumers more explicitly recognize human values and seek positive social

and environmental benefits throughout the system. As a result, LRFSs restore a sense of food citizenship among consumers. A food citizen is a resident-participant in a food system who possesses subsequent rights, duties, and responsibilities therein [102]. To foster food citizenship, the information and values flowing through a food system and its embedded values-based supply chains must be accessible to all stakeholders from producer to consumer. One aspect of restoring food citizenship is restoring confidence in credence goods in terms of quality assurance for the consumer and profitability for the producer [103]. Because information in the food supply chain is imperfect, both producers and consumers take a risk on credence goods due to customer uncertainty surrounding appropriate price values and producer uncertainty concerning tradeoffs between certification costs and price premiums [104]. One approach to build trust is to rely on regulation via third-party certification that justifies the cost of both producer compliance and consumer buy-in [105]. This type of third-party regulation necessitates a food system with a values-based supply chain, reliable data flow, and an effective labeling scheme for credence goods.

While LRFSs are expanding and replicating organically, they can be fragile systems, and little is known about their behavior at the systems level. Through a precision agroecological lens, a theoretical framework for LRFSs can be developed. Evaluation can then lead to initial design, modification, or significant reorganization in order to promote replication and durability. Precision tools accounting for variables of the social and organizational realms in which LRFSs exist may include spatial and temporal system models. Precursor diagrammatic models of food systems can identify important aspects of structure and relationships throughout the system [106]. Parameterizing models with economic, production, environmental, and social data, and simulating LRFSs, can lead to identifying the variables that influence successes and failures. Such an approach would bring a level of data-driven precision to building and managing LRFSs. Diagrammatic models and outputs from computational models can also be used as outreach tools to educate all LRFS stakeholders on system components and the flow of goods, services and information throughout. As a result, precision agroecology has the potential to restore producer consumer relationships by strengthening LRFSs, reestablishing trust in credence goods and fostering a sense of food citizenship.

## 7. Discussion

The convergence of agroecological principles and precision technology we suggest is an unusual but necessary trajectory for future farming solutions. Typically, PA falls within the production-oriented paradigm of agricultural solutions, while agroecology falls within the ecologically oriented paradigm. Though PA is often perceived as perpetuating the industrialization of agribusiness [21,107], we have shown how it can be incorporated into decision support systems parameterized with OFE to ultimately inform stakeholder-driven practices. In the same manner, agroecology has been commonly underestimated as a counterculture, low-yielding, farming movement [28,108]; however, we have shown how it is also a site-specific, quantitative science that pairs well with the management tools offered by precision technology. The merger of precision technology and agroecological principles results in a new agriculture that can be transformative by reducing inputs, substituting synthetic with sustainable inputs, incorporating more biodiversity into the system, and reconnecting producers and consumers.

Precision agroecology provides a unique opportunity to synthesize traditional knowledge and novel technology to transform food systems. In doing so, precision agroecology can offer solutions to agriculture's biggest challenges in achieving sustainability. These include environmental issues of pollution, biodiversity loss, and climate change, as well as broader societal issues of rural depopulation and corporate consolidation of the agricultural sector. Within the agroecological framework laid out earlier, tiers one, two and three tackle the prime environmental issues head on. As noted in both tiers one and two, reducing harmful agricultural inputs and substituting chemical inputs with more natural inputs, such as green manure cover crops in place of synthetic nitrogen, will reduce pollu-

tion. Synthetic nitrogen is a source of point and nonpoint pollution with cascading effects detrimental to ecological systems [109–113] and is a massive source of greenhouse gas emissions in both its production and field application [3,114,115]. Tier one can be applied to nitrogen fertilizer as a first step towards increasing efficiency gains on conventionally managed fields of farmers that are not willing to rapidly shift to substitution of inputs. Substituting nitrogen fertilizer use through well measured cover crop management, as described in tier two, pushes conventional agriculture further towards sustainability and represents the next step in shifting modern industrial agriculture to a more sustainable future. While not shown here, the concepts of precision agroecology can also reduce and replace other chemical inputs, such as pesticide applications, across the farmscape [116]. The third-tier example shows how farmscapes can be managed with precision agroecology for precision conservation of important species and prevention of biodiversity loss, while maintaining or improving agricultural output. These types of precision conservation efforts can contribute to the land sharing strategy in sustainable agriculture by providing patch habitat and ecosystem services throughout the agricultural matrix [88]. In addition to reducing greenhouse gas emissions and reducing nonpoint source pollution, precision agroecology promotes the adaptive management techniques necessary to constantly adjust to the realities of a changing climate. As farmers practice OFE by collecting and implementing their data, algorithms can be used to update best management practices. Farmers respond to greater weather uncertainty with increased purchase of crop insurance, an input to minimize risk of crop failure. However, PA data and subsequent localized crop response models can be used to quantify the risk and minimize impractical insurance costs. Furthermore, recommended variable rates of seed, fertilizer, and chemical inputs would be constantly revised based on recent climate and weather patterns. In this way, managing fields in a spatially and temporally explicit manner with precision agroecology can increase agroecosystem resiliency by confronting the realities of increasing variability and uncertainty in management outcomes which will undoubtedly increase due to climate change [39].

By transforming food systems with agroecological solutions, precision agroecology can contribute to solving broader societal issues as well. Corporate control and rising corporate profits in the farm sector have shrunk farmer profit margins and prevented small farmers from accessing the land, capital, and the technical assistance they need to succeed [117]. Precision agroecology hopes to reverse this trend through farmer empowerment. Precision agroecology promotes decision support systems for farmers to manage their own data and implement their own farm management plans. By prioritizing stakeholder engagement and empowerment, precision agroecology can avoid becoming yet another tool used by corporations to control farmers the way agrochemical inputs and genetically modified seeds have become [118]. Because precision agroecology aims to be a free technological adaptation for farmers who possess certain minimum PA technologies (which many already do) [119], its implementation will increase their net returns and improve their economic resiliency. As shown by tier four, by increasing farmer prosperity, precision agroecology can bolster producer–consumer relationships and LRFSs. Ideally, the use of precision agroecology would also promote farmer-to-farmer networks centered on knowledge exchange surrounding this novel technology [120].

Despite the best intentions of researchers and practitioners of agroecology, a fear exists that agroecology movements are being commandeered and commodified by powerful corporations [24,25,107,121]. In order to prevent the co-optation of agroecological transformation by the production-oriented paradigm, cautionary calls have been made to direct transition towards types of innovation that foster participatory processes [122,123] and safeguard the collective knowledge, rights and agency of producers [124,125]. The very issues that make PA adoption in support of agroecology challenging also provide an avenue for corporations to move in and dominate the movements. Corporations have the ability to simplify PA processes, automate them and sell the technology to farmers, thus creating a cost barrier to producer adoption. The intellectual property contained in

the algorithms, even when developed with academic institutions, should be owned by farmer cooperatives where research and development of the algorithms was cooperatively developed. Incentives need to be created for public institutions to develop precision agroecology algorithms and decision support that does not lead to intellectual property for sale to the highest industrial bidders. With this approach to precision agroecology, corporate power will be reduced, and farm efficiency gains can be passed on directly to the farmer, increasing their overall field-specific knowledge and ultimate economic and environmental sustainability.

Other barriers to adoption of sustainable agriculture include educational barriers, risk barriers, and demographic barriers [126]. For precision agroecology, these barriers refer to practices which are difficult to learn and employ (PA technology and new agroecology practices), increased risk due to uncertainty about returns on investment (time and money), and resistance to change from an older, more traditional demographic of farmers. Farmers in North America tend to be old; the average farmer in the United States is 58 years old with 92% of farmers in the United States over 35, and 34% over 65 [127]. Across all farmer demographics, farmers are less likely to experiment with new practices or technology with increasing economic and climatic uncertainty [128]. However, this resistance also provides an opportunity whereby generational shifts will inevitably occur, and younger farmers, being both more comfortable having grown up in the smartphone era and more willing to try new things, are considerably more likely to adopt new technologies [37]. Because of this, precision agroecology remains in a precarious position where uptake is low but must be readied for adoption when generational shifts inevitably occur.

Both PA and agroecology have steep learning curves which make them difficult to employ. Precision agriculture typically requires technological expertise, with devices and large data sets requiring understanding of GPS, GIS, and data management. Agroecology typically requires complex systems thinking involving plants, integrated weed and pest management, and practices such as longer rotations and cover crops. Combining these movements into precision agroecology thus inherits high barriers to adoption in terms of required new learning. In particular, the algorithms designed to wrangle large data sets and provide new management answers are sometimes black boxes even to data scientists, and farmers should not be expected to master advanced statistics. However, through effective communication and well developed and automated but interactive decision support systems, the process of precision agroecology can eventually be made both user-friendly and empowering for the farmer with clearly presented findings. Analysis of the data and algorithms needs to be open-source and designed to be interactive with the farmer to gain insights into the complex ecological processes that can result in non-intuitive outcomes of management actions. In this way precision agroecology can augment farmer knowledge, rather than replace it, and thereby become a trusted and powerful tool by farmers who adopt it [129–131]. We therefore highlight the importance of designing an approachable interface between data collection and decision-makers, further facilitated by designing applications based on free, open-source data and interactive analysis.

## 8. Research Gaps and Future Research Directions

To breach the current research gap, farmers need decision support systems to distill the information and data gathered from farms and OFE to inform management. Development of these systems is of utmost importance for the adoption of precision agroecology [132]. While PA technology makes it easy to obtain large quantities of site-specific data for producers, decision support tools are necessary to implement data-driven management [133]. Start-ups and corporations have been developing decision support systems, such as Adapt-N, FieldNETAdvisorTM, FarmBot, Climate Corporation, FaunaPhotonics and Field to Market to relay PA data to user-friendly formats with the intent to guide sustainable management [134,135]. In response to Ingram & Mayes' [120] recent call for

co-created digital technologies that prioritize a user-centric approach, our lab is working to create adaptive management tools that incorporate both big data and producer knowledge through simulation and structured decision making, such as through the 'OFPE' R package (https://github.com/paulhegedus/OFPE.git (accessed on 17 December 2021)) and the On-Farm Experimentation Prescription Generator (http://trialdesign.difm-cig.org/home (accessed on 10 December 2021)). These tools aim to empower farmers to control and use their own or open-source data, sidestepping corporate middlemen, and thereby retaining decision making processes on-farms.

However, one limitation of this study is that while results and recommendations from OFE are inherently "black box" and enigmatic, they must also be practical and applied on farms to retain their value. The conundrum of OFE is that big data requires advanced analysis to assess a multitude of complex on-farm interactions and yet must remain transparent, inclusive of farmer knowledge and easy to apply. However, a vast research gap currently surrounds stakeholder attitudes towards the adoption of precision agroecology due to uncertainty about data availability, usability and security. This uncertainty underlies the main limitation of this study, which is the lack of trust between the technology industry and farmer stakeholders. This barrier to trust will undoubtedly limit the adoption of the very precision technologies that OFE relies on. Due to the fact that PA is at the crux of an intellectual property battle, this study was limited in its ability to showcase a number of suppressed, small-scale efforts to develop open-source decision support tools and likely overlooked a number of current attempts to do so. Future research and development of user-friendly PA technology is paramount to creating a more equitable and sustainable food system.

Moving forward, precision agroecology can address tier five of agroecology by creating a just and equitable global food system. To transform the global food system, policy surrounding agriculture needs to be data driven [136]. Policy makers should have access to the stream of data from PA to create incentives and regulations that account for the environmental and social relationships surrounding agriculture and food systems. Specifically, precision agroecology lends itself to shifting the focus of agricultural systems to food and environmental quality rather than quantity of food production. Utilizing precision agroecology can provide a unique opportunity to improve agriculture's impact on human health, an aspect of the social relationships surrounding food systems. Rather than externalizing environmental and human health costs like the industrialized agricultural sector, precision agroecology can price in negative externalities by providing data, derived from PA technology, to support policy that properly pays for food quality. Future research and policy should prioritize crop quality over quantity and incentivize producers and markets to manage and price crops for their quality in terms of nutrient content. In addition, reframing agriculture with a focus on environmental quality would incentivize best management practices such as reimbursing producers for optimizing ecosystem services. By prioritizing both food and environmental quality, precision agroecology can restore the values of nature and nutrition over production and profit to rebalance agri-food systems in a sustainable and resilient manner.

## 9. Conclusions

Precision agroecology offers solutions to the problems faced by modern industrial agriculture by utilizing the technologies of industrial agriculture to inform agroecological decisions. Agriculture is one of the largest global markets and change is unlikely to occur quickly. Adapting agroecological philosophies in policy and farmer decisions will require concerted and coordinated efforts at all scales for which the tiers of agroecology span. Precision agroecology shifts the paradigm of agricultural systems towards a more sustainable future by harnessing the technologies and data rapidly developed and generated from industrial management practices like PA. Precision agroecology serves as a compromise between the divergent factions of agriculture and bridges the gap between seemingly opposite ideologies through the use of data and analytics. Precision agroecol-



ogy increases efficiencies of farms, and with further OFE and policy incentives can lead to data-informed substitution of inputs, conservation of uncropped areas to maximize ecological benefits, and reestablishment of direct relationships between producers and consumers. The data gathered from precision agroecological management thus provides a resource for informing policy decisions at all scales to offer transformative agri-food systems solutions that were not possible previously. Therefore, we propose precision agroecology as an effective and necessary trajectory towards future farm sustainability. As agriculture develops in the age of climate awareness and technological advancement, precision agroecology offers an opportunity to transition agriculture towards agroecological principles.

**Author Contributions:** Conceptualization H.D., P.B.H. and S.L.; writing—original draft preparation, H.D., P.B.H. and S.L.; writing—review and editing, H.D., P.B.H., S.L., T.B. and B.D.M.; visualization, H.D. and S.L.; supervision, B.D.M.; project administration, H.D.; funding acquisition H.D., P.B.H., S.L. and B.D.M. All authors have read and agreed to the published version of the manuscript.

**Funding:** This research was funded by Western Sustainable Agriculture Research and Education, grant numbers GW 19-190, GW 19-198 and GW19-199. This research was also funded by a USDA-NIFA-AFRI Food Security Program Coordinated Agricultural Project, titled "Using Precision Technology in On-farm Field Trials to Enable Data-Intensive Fertilizer Management", (Accession Number 2016-68004-24769), and also by the USDA-NRCS Conservation Innovation Grant from the On-farm Trials Program, titled "Improving the Economic and Ecological Sustainability of US Crop Production through On-Farm Precision Experimentation" (Award Number NR213A7500013G021). Grants from the Montana Fertilizer Advisory Council 2016–2021.

**Institutional Review Board Statement:** Not applicable.

**Informed Consent Statement:** Not applicable.

**Data Availability Statement:** Not applicable.

**Acknowledgments:** We would like to thank our farmer collaborators who have shaped our perspectives and shared their land and knowledge with us.

**Conflicts of Interest:** The authors declare no conflict of interest.

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
