# Peer review of "Precision Agroecology"

_sustainability, doi:10.3390/su14010106_

Round 1
Reviewer 1 Report
After reading this work, I have few questions need to be answered:
(1) Normally, we have 5 keywords in maximum. Please reduce keywords at appropriate level.
(2) In paper cited references, some references based on the alphabetical order, while others based on the year order. Please check the whole paper for consistency purpose.
(3) This paper is a review paper, what are the research gaps emerged from this study and corresponding future research directions?
(4) What are the limitations of this study?
(5) Where is the research methodology section?
I think this paper contributes to the literature and knowledge significantly. Thus, minor revision is appropriate.
Author Response
Please see the attached response to your comments.

Reviewer 2 Report
Dear Editorial Board, Dear Authors,
The paper entitled “Precision Agroecology” aims to study the possibility of uniting these two seemingly diverging paradigms of production oriented and ecologically oriented agriculture in the form of precision agroecology.
The paper raises an important issue about precision agroecology can offer solutions to agriculture’s biggest challenges in achieving sustainability in a major state of global change.
The paper poses the research question: is precision agroecology the future of agriculture?
After reading the paper, I have comments and suggestions to improve the paper as follows:
The paper does not contain a chapter Materials and Methods.The presentation and description of the research aprocedure is missing. There is no figure that presents the research procedure. The description is confusing and difficult to understand.
The Results were presented and described in a very good manner and are very interesting. They contribute to the value of this paper. No changes are required.
In the Discussion Section, the authors should discuss and explain the findings and results of the paper more. It also important to describe the results of the paper in greater detail in this section. This would contribute to a high improvement of this paper. The authors should compare their project and results with results from similar conducted research on this topic from other the world.
In paper there were many technical errors that need to be removed:
- badbibliography notation- nconsistent with the guidelines of Sustainable
- incorrect record of literature and sources in the text (should be numbers and not names).
- the Figures need sources.
Furthermore, I strongly recommend, to revise and check the language, grammar and spelling in the paper.
Kind regards,
Author Response
Please see the attached point-by-point response to your comments.

Reviewer 3 Report
I would like to thank the authors for an interesting topic. The authors explained the possibility of uniting two paradigms of production oriented and ecologically oriented agriculture in a very clear and easy to understand way.
I have just a suggestion to avoid using questions in the text. Moreover, Table 1 should have its title, not the full sentence, because the explanation of the table should be in the text above..
Author Response

(The authors gave the same response as above.)
